# Possible Adverse Effects of Food Additive E171 (Titanium Dioxide) Related to Particle Specific Human Toxicity, Including the Immune System

**DOI:** 10.3390/ijms22010207

**Published:** 2020-12-28

**Authors:** Nicolaj S. Bischoff, Theo M. de Kok, Dick T.H.M. Sijm, Simone G. van Breda, Jacco J. Briedé, Jacqueline J.M. Castenmiller, Antoon Opperhuizen, Yolanda I. Chirino, Hubert Dirven, David Gott, Eric Houdeau, Agnes G. Oomen, Morten Poulsen, Gerhard Rogler, Henk van Loveren

**Affiliations:** 1Department of Toxicogenomics, GROW-School for Oncology and Developmental Biology, Maastricht University Medical Center, P.O. Box 616, 6200 MD Maastricht, The Netherlands; t.dekok@maastrichtuniversity.nl (T.M.d.K.); dick.sijm@maastrichtuniversity.nl (D.T.H.M.S.); s.vanbreda@maastrichtuniversity.nl (S.G.v.B.); j.briede@maastrichtuniversity.nl (J.J.B.); 2Netherlands Food and Consumer Product Safety Authority, P.O. Box 43006, 3540 AA Utrecht, The Netherlands; j.j.m.castenmiller@nvwa.nl (J.J.M.C.); a.opperhuizen@nvwa.nl (A.O.); henk.van.loveren@gmail.com (H.v.L.); 3Unidad de Biomedicina, Facultad de Estudios Superiores Iztacala, Universidad Nacional Autonóma de México, Mexico City 54090, Mexico; irasemachirino@gmail.com; 4Norwegian Institute of Public Health, P.O. Box 222 Skøyen, 0213 Oslo, Norway; hubert.Dirven@fhi.no; 5Food Standard Agency, London SW1H9EX, UK; dmgott@btinternet.com; 6French National Research Institute for Agriculture, Food and Environment (INRAE), 75338 Paris, France; eric.houdeau@inrae.fr; 7National Institute for Public Health and the Environment, P.O. Box 1, 3720 BA Bilthoven, The Netherlands; agnes.oomen@rivm.nl; 8National Food Institute, Technical University of Denmark, 2800 Kgs. Lyngby, Denmark; morp@food.dtu.dk; 9Department of Gastroenterology and Hepatology, University Hospital of Zurich, 8091 Zurich, Switzerland; Gerhard.Rogler@usz.ch

**Keywords:** titanium dioxide, TiO_2_, E171, food additive, food safety, nanomaterial, nano size, oral exposure, mode of action, adverse health effects, toxicity, review

## Abstract

Titanium dioxide (TiO_2_) is used as a food additive (E171) and can be found in sauces, icings, and chewing gums, as well as in personal care products such as toothpaste and pharmaceutical tablets. Along with the ubiquitous presence of TiO_2_ and recent insights into its potentially hazardous properties, there are concerns about its application in commercially available products. Especially the nano-sized particle fraction (<100 nm) of TiO_2_ warrants a more detailed evaluation of potential adverse health effects after ingestion. A workshop organized by the Dutch Office for Risk Assessment and Research (BuRO) identified uncertainties and knowledge gaps regarding the gastrointestinal absorption of TiO_2_, its distribution, the potential for accumulation, and induction of adverse health effects such as inflammation, DNA damage, and tumor promotion. This review aims to identify and evaluate recent toxicological studies on food-grade TiO_2_ and nano-sized TiO_2_ in ex-vivo, in-vitro, and in-vivo experiments along the gastrointestinal route, and to postulate an Adverse Outcome Pathway (AOP) following ingestion. Additionally, this review summarizes recommendations and outcomes of the expert meeting held by the BuRO in 2018, in order to contribute to the hazard identification and risk assessment process of ingested TiO_2_.

## 1. Background of TiO_2_ as a Food Additive

Titanium dioxide (TiO_2_) is a widely used white pigment and opacifying agent, with applications in paints, pharmaceuticals, cosmetics, and food [1]. When used as a food additive in the European Union (EU), it is listed as E171 to refer to a specified food-grade form of TiO_2_, which has no nutritional value and is used to attain a white color, shade other pigments, or in pharmaceuticals [2]. The whitening is best achieved with TiO_2_ particles within a size range of 200–300 nm, due to their light scattering effects [3]. TiO_2_ occurs in nature in three distinct crystal structures—anatase, rutile, and brookite, but only anatase and rutile are allowed as a food additive [4,5,6]. The European Union allows E171 (anatase and rutile in uncoated, no surface treatment forms) in quantum satis (without limitations), based on its low absorption and subsequent low toxicity, presumed inertness, and low solubility [5,7,8]. Its low toxicity and inertness, however, are being debated, as long-term inhalation studies over two years have shown the development of lung tumors in rats, following exposure to high concentrations of TiO_2_ [9,10]. As a consequence of these findings, the International Agency for Research and Cancer (IARC) has classified TiO_2_ as “possibly carcinogenic to humans after inhalation” [10]. In 2017 the Risk Assessment Committee (RAC) of the European Chemical Agency (ECHA) published an opinion that proposed the classification of TiO_2_ as a category 2 carcinogen after inhalation, according to the criteria of the Classification, Labelling and Packaging (CLP) Regulation [11]. On the 18 February 2020, the EU took over ECHA’s opinion and published the classification of TiO_2_ as a suspected carcinogen (category 2) by inhalation in powder form with at least 1% particles with aerodynamic diameter ≤ 10 μm, under the CLP Regulation (EC No 1272/2008). The classification will apply on 1 October 2021 after an 18-month transition period [12]. What the observed toxicity and hazard classification following inhalation mean for oral toxicity is of yet not clear.

Over the last years, an increasing number of studies investigated the behavior and effects of E171 and nano-sized TiO_2_ after ingestion and discovered potential adverse effects, including the induction of inflammation, the formation of reactive oxygen species (ROS), and co-genotoxic effects [13]. Sub-acute and sub-chronic studies also revealed the induction of epithelial hyperplasia and preneoplastic lesions in the colon of rats and mice after the ingestion of E171, while other oral toxicological studies did not confirm such effects [14,15,16,17,18]. For the oral intake of food additive E171, the European Commission requested a re-assessment of TiO_2_ by the European Food Safety Authority (EFSA), following the publication of studies by ANSES in 2017. EFSA concluded that the results of these studies did not merit a re-opening of the existing opinion but suggested to fill in the existing data gaps, reduce uncertainties and evaluate new findings carefully in regard to their adverse effects and physicochemical properties of the TiO_2_ particles used [7,8,19,20,21]. The re-assessment of TiO_2_ has recently been opened and was initiated in 2020 by the EFSA [22].

Parallel to the EFSA activities, the Office of Risk Assessment and Research (BuRO) at the Netherlands Food and Consumer Product Safety Authority (NVWA) organized a workshop that was held in July 2018, regarding the “potential health effects of the food additive titanium dioxide (E171)”, on which BuRO based its opinion that was published in 2019 [23].

In response to signals in the scientific literature about potentially harmful effects after ingestion of E171 in rodents and the widespread use of this substance in foods, BuRO identified the following questions in the process of risk assessment of E171 that have to be addressed:
Does oral exposure to E171 or nano-sized TiO_2_ reveal a relevant toxicological hazard?How reliable are these in-vitro and in-vivo studies?Are the animal models, the exposure conditions, and the effects observed in these studies relevant to humans?Can the data from in-vitro and in-vivo studies with TiO_2_ be extrapolated to humans?Are there epidemiological studies on the effects of E171 in humans after oral exposure?

Since this workshop in 2018, more studies have been published that investigated the concerns of adverse effects arising from E171 ingestion. This literature review integrates the main conclusions of the expert meeting initiated by BuRO, with recently published studies in order to present an overview of relevant findings regarding E171 toxicity after oral intake. The literature search on PubMed and EmBase was conducted from June 2020–September 2020 and included the search criteria “TiO_2_”, “titanium dioxide”, “E171” with publication dates from 2018–2020. Previous scientific papers in the field, as well as references in these publications were evaluated. The present literature review aims to shed light on the importance of complete particle characterization, on the effects of matrices, and highlight toxicological relevant pathways potentially involved in the induction of adverse health effects following E171 ingestion. Additionally, it provides approaches to decrease uncertainties concerning the health effects of E171 consumption, and finally formulates recommendations for future studies and follow-up actions regarding the risk assessment of E171.

## 2. Physicochemical Properties and Characterization of E171

Titanium is one of the most abundant elements in the earth’s crust, which occurs in nature only in its oxidized form as titanium dioxide or Ti(IV) oxide. Once processed, TiO_2_ is a white, odorless powder that is poorly soluble in aqueous solutions [2,5]. The anatase form TiO_2_ is most frequently used as a whitening agent in foodstuff, despite its high surface reactivity and ability to generate ROS in an aqueous solution after UV irradiation [2,24]. Food-grade TiO_2_/E171 consists of micro-and nanoparticles with a primary particle size ranging from 60–300 nm [25]. Around 10–40% of the pristine TiO_2_ particles in E171 are estimated to be smaller than 100 nm and can therefore be considered as nanoparticles [25,26,27,28]. However, according to the Commission’s recommendation in 2011 (2011/696/EU), a nanomaterial must contain over 50% of nanoparticles, which excludes E171 of this category [28,29]. Based on information reported in the literature the EFSA Panel on Food Additives and Nutrient Sources suggest that the food additive E171 mainly consists of micronized TiO_2_ particles ranging from 104–166 nm and a percentage of particles < 100 nm ranging from 5.4–45.6% [21,30].

Recently published work by Verleysen et al. (2020) showed that 12 out of 15 pristine E171 materials purchased from manufacturers consist of more than 50% TiO_2_ particles that are smaller than 100 nm and that commercially available anatase E171 materials constitute of 18–74% (TEM) or 32–64% (sp-ICP-MS) nanoparticles [30]. This examination assigns a larger fraction of TiO_2_ particles present in pristine E171 to the nano-sized fractions than previously assumed. Analysis of food samples containing E171, via ICP-MS and Raman spectroscopy, showed anatase type TiO_2_ particles in the range of 26.9–463.2 nm, with 21.3–53.7% of the particles in the nano-size fraction [31]. The determination of the nanoparticle fraction (Figure 1) within E171 is of importance since the size of particles is considered to be an important factor influencing toxicokinetics, toxicodynamics, and thus toxicity [8,21,32]. Nanoparticles display a higher surface to volume reactivity, translocation properties, bioavailability, and increased cellular interactions than larger particles [33].

The shape, size, and state of agglomeration and aggregation are important properties regarding the effects of food-grade TiO_2_. Generally, it is assumed that the round and spherical crystal forms of TiO_2_ contribute to a lower extent to the induction of adverse effects, when ingested [34]. The size of food-grade TiO_2_ particles, on the other hand, plays an important role regarding their toxicity. Nano-sized TiO_2_ particles are suspected to induce more adverse effects, including ROS formation, cytotoxicity, and increased release of inflammatory cytokines, compared to micro-sized TiO_2_ particles [32,35,36]. Proquin et al. (2018) demonstrated that a mixture of nano- and micro-sized TiO_2_ particles, as they are present in E171, induce more adverse effects than the single fractions alone. This emphasizes the importance of testing food-grade TiO_2_ particles as a whole, rather than its nano- and micro-sized fraction [16].

The interaction of E171 with its direct environment and colloidal stability are other factors that need to be considered during its characterization [37]. Suspended TiO_2_ particles tend to agglomerate or aggregate, according to their isoelectric point and the pH of the milieu, leading to the formation of larger clusters. Aggregation describes the assembly of primary particles through covalent or metallic bindings, while agglomeration results from van-der-Waals interactions, hydrogen bonds, adhesion by surface tension, or electrostatic attraction [2,38]. The determination of agglomeration and aggregation status is crucial because it can significantly alter hydrodynamic diameter, size, and the stability of particle-complexes, thus affecting uptake, reactivity, and toxicity [39].

The high surface area, charge, and chemical properties of TiO_2_ particles provide the possibility of many biomolecules to be adsorbed. The formation of a protein corona can change the physicochemical properties of TiO_2_ particles, e.g., their reactivity and the interactions of these particles with their environment, including cellular uptake, accumulation, intracellular localization, distribution, and release [40]. The variability of protein coronas is dependent on the different molecules present at each location and can influence their interaction with cells [41]. The presence of transferrin in the protein corona, for example, can affect the clathrin-mediated endocytosis via the transferrin-receptor and result in significantly altered particle internalization [42].

The formation of protein coronas can also lead to conformational changes of the proteins themselves, resulting in irreversible changes to secondary protein structures and leading to protein dysfunction [43]. Additional interactions of TiO_2_ nanoparticles with non-protein components might be harmful too. Bianchi et al. (2017) showed that the endotoxic effect of lipopolysaccharides (LPS) is increased when bound to TiO_2_ nanoparticles, resulting in the potentiation of pro-inflammatory effects including induced expression of nuclear factor kappa-light-chain-enhancer of activated B-cells (NF-kB) and interferon regulatory factor 3 (IRF-3) dependent cytokines [44]. The consideration of TiO_2_-protein-corona-complexes in the characterization and determination of physiochemical properties and adverse effects of food-grade TiO_2_ is important for an adequate safety evaluation.

For this reason, it is important to carefully examine and analyze the physicochemical characteristics of TiO_2_ particles in its vehicle, as well as in its surrounding matrix as their final milieu, to guarantee a profound assessment of potential adverse health effects of E171 and to adequately compare different studies in the process of risk assessment.

## 3. Exposure to E171

E171 is used in products such as candy, coffee creamer, chewing gum, sauces, nutritional supplements, toothpaste, and pharmaceuticals. Although both the anatase and rutile forms of TiO_2_ are authorized for foods, the characterization of European and American food samples showed that anatase is the predominant TiO_2_ crystalline structure used as food additive E171 and thus the main source of exposure for the general population [27,45,46,47,48]. The intake of E171 varies between different age groups and countries, while children, in general are the most highly exposed group, due to their lower body mass and disproportionally higher consumption of E171-containing products [5,49]. Table 1 shows the estimated daily intake of E171 per kg body weight (bw) in different countries and age-groups [49,50].

The highest concentrations of E171 are found in chewing gum, candies, and powder sugar toppings such as icings. Chewing gums contain between 1.1 mg (±0.3 mg) to 17.3 mg (±0.9 mg) TiO_2_ particles per piece of gum with a mean average weight per piece of 1416 mg (±27 mg) to 2240 mg (±86 mg) [26]. TiO_2_ nanoparticles account for up to 19% (±4) of all particles present in these gums [26]. The accidental ingestion of toothpaste, while brushing teeth is another major source of E171 intake, that can result in an exposure of 0.15 to 3.9 mg/day, when 10% of toothpaste is ingested [50]. Additional release of TiO_2_ particles (70–200 nm) from food packaging materials or food-related products, such as frying pans, may also contribute to TiO_2_ ingestion [55]. The focus of oral TiO_2_ exposure estimation should potentially be extended from the food additive E171 to personal care products, packaging, and coating of household items [28,33,55]. Daily dietary intake of E171 can reach several hundred milligrams, of which at least 10–40% are in the form of TiO_2_ nanoparticles. The long-term exposure to such quantities of nano- and micro-sized TiO_2_ raises concerns about the risk of potential accumulation in organs and potentially harmful effects on human health [27].

## 4. Toxicokinetics of Ingested E171

Following the oral ingestion of E171, the key question is how much of the E171 and which portion of each size fraction will be absorbed along the oro-gastrointestinal route before it is exerting local effects. E171 is expected to be systemically distributed via the blood circulation or the lymphatic system to various organs and tissues. Digestive enzymes and pH levels in the mouth, stomach, and small intestine may alter and change physicochemical properties of E171, including the protein corona formation, and therefore have the potential to affect the absorption in vivo.

The absorption of different sized TiO_2_ particles (148, 36, 28 nm) in a porcine buccal model showed that all investigated particles permeate the mucosa layer and enter the oral epithelium. Penetration depth varied with particle size, with smaller particles penetrating deeper. This ex-vivo model demonstrated that TiO_2_ particles can also enter the buccal mucosa under physiological conditions, which included digestive enzymes e.g., mucins, and relevant pH levels [56,57]. Absorption and internalization of E171 have been studied in human epithelial colorectal adenocarcinoma cells (Caco-2 cells) and various other in vitro models. Nanosized TiO_2_ particles were effectively entrapped by Caco-2 cell monolayers and stored in the affected enterocytes. The internalized TiO_2_ nanoparticles showed a tendency to agglomerate or aggregate in the cytosol, but nanoparticles and enveloped nanoparticles in cytoplasmic vesicles could also be observed. This internalization of TiO_2_ nanoparticles in differentiated Caco-2 cell monolayers after 4 h of exposure indicates a transcellular absorption [58]. The exposure of differentiated Caco-2 cell monolayers to TiO_2_ nanoparticles resulted in increased epithelial permeability, indicating a disruption of the cytoskeletal integrity, increased tight-junction (TJ) permeability, and downregulation of genes encoding for tight junction proteins [58,59].

In vivo, TiO_2_ can cross the regular ileum and follicle-associated epithelium before it translocates and enters the Peyer’s patch in the colon [60]. Comera et al. (2020) recently showed that TiO_2_ is mainly taken up by crossing the regular epithelium of the small bowel villi. This process is facilitated by goblet cell-associated passage and passive diffusion through the paracellular tight junction spaces between the enterocytes, without displaying epithelial transcytosis patterns [61]. This indicates that the translocation of TiO_2_ nanoparticles in the ileum is mainly facilitated through paracellular resorption, transepithelial absorption, and potentially through the impairment of paracellular junctions [58,59,60]. Studies in rats showed that only a very small fraction of 0.007 to 0.6% ingested E171 is absorbed and enters the circulation [18,62]. These observations are consistent through various species, including rats, mice, and *Drosophila melanogaster* [14,63,64]. When TiO_2_ nanoparticles (25 or 75 nm) enter the circulation, in laboratory rats, they deposit in the liver and spleen, where they exhibit a half-life time of more than 30 days, resulting in a high risk of bioaccumulation given the chronic daily exposure [18,62,65]. Increased TiO_2_ tissue levels have been found in the spleen and ovaries of rats, along with sex-related histological changes in the thyroid, adrenal medulla and adrenal cortex (female) and thyroid function (male). These findings indicate the possibility of endocrine and reprotoxic effects after the ingestion of E171 [66]. While a majority of the published oral in vivo studies identify a minor absorption of E171 in rats and mice, others showed that certain forms of TiO_2_ (rutile) did not migrate from the gastrointestinal tract [18,67].

Studies in humans on orally administrated TiO_2_ showed a low bioavailability [68,69,70]. Basal titanium blood levels ranged between 5.9–18.1 µg/L (mean 11.1 µg/L) and peaked after 8–12 h at 37.4–49.7 µg/L after ingestion of 22.9 mg TiO_2_ in a gelatin capsule. Administration of 380 nm sized TiO_2_ (anatase) showed lower absorption than 160 nm sized TiO_2_ (anatase). The highest titanium blood concentration was detected at 109.9 µg/L, after the ingestion of 45.8 mg TiO_2_ in a gelatin capsule, after 8 h, showing large differences in absorption among the group of six male volunteers [68]. The ingestion of 100 mg food-grade TiO_2_ (E171) increased total titanium blood levels after 6–8 h, with peak titanium blood concentrations reaching 10 ppb in comparison to 1.5 ppb basal levels [70]. Contrary to these findings, the study of Jones et al. (2015), which used different sized TiO_2_, showed no statistically significant absorption of TiO_2_, after the ingestion of 5 mg/kg bw TiO_2_ [69]. Even though the absorption of ingested TiO_2_ over a healthy intestinal barrier seems to be very low, it is important to take into consideration factors like net volume of translocated particles through the gut barrier, possibly impaired intestinal barrier function that facilitates TiO_2_ particles translocation and bioaccumulation in systemic organs, when accurately assessing potential health hazards.

Heringa et al. (2016) and Rompelberg et al. (2016) published an overview of studies examining the absorption of ingested TiO_2_ nanoparticles [1,49]. Following their physiologically based pharmacokinetic (PBPK) modeling, these researchers concluded that TiO_2_ nanoparticles can be absorbed, although at a very small rate of approximately 0.02 to 0.05% [51,71,72,73]. The translocation into lymphatic and blood circulation can lead to the deposition of TiO_2_ nanoparticles within tissues and organs after ingestion [71]. The deposition of TiO_2_ in humans was observed in the Peyer’s patch, especially in patients suffering from inflammatory bowel diseases (IBD) [74,75].

Based on the oral exposure estimation of the Dutch population, using external dosage, no risk of adverse effects is expected in humans, except potential effects on ovaries [1]. However, if toxicokinetic information based on internal organ concentration and accumulation over time of TiO_2_ nanoparticles was included, the potential additional risk for liver and testis was identified [1]. Additional work of Heringa et al. (2018) and Brand et al. (2020) showed that post-mortem collected human liver (median 0.03 mg/kg), jejunum (median 0.14 mg/kg), ileum (median (0.26 mg/kg), kidney (median 0.06 mg/kg) and spleen (median 0.04 mg/kg) contain titanium particles and that they accumulated both micro-and nanosized TiO_2_ [13,76]. The quantities detected in these organs were partially higher than levels that are considered safe for humans, after applying conventional safety factors [76]. ICP-MS and TEM-EDX analysis of human placentae and meconium (first stool of newborns) collected from normal pregnancies suggest a maternofetal passage of TiO_2_, which does not provide information on the source of TiO_2_ particles in these organs and routs for maternal exposure. However, placenta perfusion experiments with E171 suspension confirmed a low transfer of food-grade TiO_2_ particles to the fetal side. The diameter of the TiO_2_ particles recovered in the fetal exudate showed that 70 to 100% of particles were in the nanosized range [77]. These findings suggest that the human placenta barrier is not able to completely prevent the passage of TiO_2_ nanoparticles to the fetus and emphasized the need to assess the risk of TiO_2_ nanoparticles during pregnancy [77].

Independent from the extent of TiO_2_ absorption, a considerable amount of TiO_2_ (approximately 99%) is retained and accumulated in the intestinal lumen, before it is excreted via the feces, without undergoing any alteration or metabolization [78,79]. Due to accumulation in the intestinal lumen before excretion, interactions of TiO_2_ with the gut microbiota are possible, which may lead to a modification of intestinal homeostasis and which could possibly impact the health of the host [80].

Uncertainties remain concerning possible effects of the food matrix, on E171 absorption, distribution, metabolism, and excretion. The matrix can potentially alter the physicochemical properties of E171 substantially and influence the degree of absorption. The various influences of digestive processes through saliva, stomach acid, and intestinal pH on E171, its protein corona and its physicochemical properties, bioavailability, and potential adverse effects, are currently poorly understood.

## 5. Health Effect of Ingested E171

Potential health risks resulting from the ingestion of E171 are still under discussion. Here, an overview of in-vivo, in-vitro, and ex-vivo toxicity studies with TiO_2_ nanoparticles and food-grade E171 is provided.

### 5.1. In Vivo Toxicity of E171

In 1979, NTP concluded that the in vivo carcinogenicity studies in rats and mice, they performed demonstrated that TiO_2_ can be considered as safe as a food additive [81]. These studies were carried out in *Fisher 344* rats (male 1125/2250 mg/kg/bw and female 1450/2900 mg/kg/bw/day) and *B6C3F1* mice (male 3250/6500 kg/bw/day and female 4175/8350 mg/kg/bw/day) via daily dietary administration of pigment grade anatase [2,81] Microscopical images suggested a mean particle diameter between 200–300 nm, but no specific size characterization was conducted. The test material was included in the diet of the mice, without considerable effects on the survival of male mice. The female mice in the highest dose group, showed a survival rate of 66% at the end of the 104-weeks study, in comparison to 90% survival in the control group. Histopathological examination showed a dose-dependent increase in hepatocellular carcinoma in male mice from 17% in the control group to 29% in the high-dose group. These effects remained in the range of historical control data. Histopathological examination showed an increase in hyperplastic bile ducts in male rats in low- and high-dose groups after 103-weeks of exposure. Female rats showed an overall increased incidence in c-cell adenomas and carcinomas of the thyroid from 2% in the control to 14% in the high-dose group. No adenomas or carcinomas have been detected in the low-dose group. The statistical analysis led to the conclusion that the incidence is not statistically significant nevertheless does the occurrence of thyroid tumors need to be carefully considered [2,81,82].

Table 2 shows an overview of recent in vivo studies, assessing various adverse health effects of TiO_2_ nanoparticles and E171 following ingestions e.g., genotoxic effects, inflammation, oxidant-antioxidant-balance, and mortality. Some studies with E171 and TiO_2_ nanoparticles showed no adverse effects, even at extremely high doses of up to 24,000 mg/kg bw/day [18,67,83,84]. Dietary administration of E171 in rats for 7 and 100 days showed no effect on histopathology of the small and large intestine, liver, spleen, lungs, or testes and no effects on aberrant crypt formation, goblet cell number, or colonic gland length. Administration of E171 via the food did not result in any effects on immune parameters, including interleukins, INF-γ, or TNF-α, nor tissue morphological changes [17]. Other studies in rats and mice showed intestinal inflammation, hepatotoxic effects, changes in levels of alkaline phosphatase (ALP), alanine aminotransferase (ALT), aspartate aminotransferase (ASP), and effects on oxidants and antioxidants, including reduced and oxidized forms of glutathione (GSH/GSSG), glutathione peroxidase (GPx), superoxide dismutase (SOD), and catalase (CAT) [14,35,85,86,87,88,89]. Intestinal inflammation was often accompanied by alterations in gene expression and activity TNF-α, IFN-γ, IL-2, IL-8, IL-10, NF-kB, cytochrome p450 (CYP450), cyclooxygenase-2 (COX-2), Ki67, and T-helper cells 1 (Th-1) [15,86,88,90]. Some studies reported increased genotoxicity in the form of DNA damage, micronuclei, and dysplastic alterations of tissues including the distal colon and liver, while others did not [15,35,90,91,92,93,94]. Markers indicating the progression of tumors (COX-2, Ki68, p65, TNF-α, β-catenin), alteration of tumor-related pathways mitogen-activated protein kinase (MAPK) and olfactory/G-protein-coupled receptor family (GPCR) have been measured [15,88,92,95]. Changes in metabolic function, telomere shortening, *TJP-1* gene expression, insulin resistance, endoplasmic reticulum (ER) stress, impaired cell cycle, and increased mitotic indices are other reported adverse effects [85,88,91,93,94,95]. Additional studies in vivo and ex vivo following i.v. and i.p. injections and ex vivo experiments in rats and mice, allowing for higher systemic absorption, confirm genotoxic effects e.g., formation of micronuclei (bone marrow), inflammatory responses in the liver, and secretion of IL-β in bone marrow-derived macrophages from mice [96,97,98].

There is increasing evidence that the exposure of E171 can alter gut microbiota in laboratory animals, resulting in changes of colonic pH and abundance of certain commensal bacteria, which in turn can result in increasing levels of LPS, potentially increasing lipid peroxidation processes and a significant increase in oxidative stress [85]. Dietary exposure to E171 has been linked to effects on gut microbiota and intestinal health in experimental animals, where even low exposure interferes with the gut microbiome, causing low-grade intestinal inflammation and exacerbating existing intestinal health conditions [85,100,101,102,103]. Decreased crypt length, infiltration of CD8+ cells, and macrophages, as well as increased expression of inflammatory cytokines, indicate impacts on gut homeostasis and colonic inflammation in vivo [104]. Alterations of the microbiota-immune axis have been associated with IBD, metabolic disorder, and colorectal cancer (CRC) [105,106,107]. Chronic exposure to TiO_2_ and its effects on intestinal health, especially in relationship with impaired intestinal barrier function, seen as in IBD patients and their potential risk of increased TiO_2_ absorption due to their impaired intestinal barrier integrity, have to be carefully investigated since they may represent a population with a higher risk of E171 related adverse health effects [80].

TiO_2_ nanoparticles can cross the blood-brain-barrier (BBB) in rats and mice and accumulate in the brain, leading to an increase of oxidative stress and nitric oxide (NO) levels. TiO_2_ accumulation leads to histopathological changes of the brain, inflammation, decreased acetylcholinesterase levels, decreased expression of inflammatory markers such as TNF-⍺, IL-6, and GSH depletion. TiO_2_ toxicity on the brain might increase the risk of Parkinson’s disease, through the destruction of dopaminergic neurons [108,109,110].

Examination of cardiotoxic effects of E171 and TiO_2_ nanoparticles revealed effects on vasomotor function, including the increase of acetylcholine-induced vasorelaxation, serotonin-induced vasoconstriction, and nitroglycerin levels [111,112].

Reprotoxic and developmental toxic effects have been shown for TiO_2_ nanoparticles. Exposure decreased testis weight, serum testosterone levels, and induced histopathological changes and anomalies in the sperm of mice [113]. Pregnant mice, which have been exposed to E171, showed altered gene expression related to apoptosis, brain development, and oxidative stress in their newborn pups [114]. The intragastric administration of TiO_2_ nanoparticles in rats showed an increase in gamma-glutamyltransferase (gamma-GT), decreased testicular steroidogenic regulatory protein (StAR), *c-kit* gene expression, serum testosterone level, and sperm count. Exposed animals also exhibited prostatic and testicular altered GSH levels, elevated TNF-⍺ concentration, up-regulated *Bax*, *Fas*, and *caspase-3* gene expression, downregulation of *B-cell lymphoma-2* (*BCL-2*) gene expression and enhanced prostatic lipid peroxidation. Sperm malformation elevated testicular acid phosphatase activity and MAD levels, serum prostatic acid phosphatase activity, prostate-specific antigen (PSA), gonadotrophin, and estradiol levels occurred after 2 and 3 weeks administration [115]. Chronic TiO_2_ nanoparticle exposure in zebrafish showed a significant impairment of their reproduction, resulting in reduced numbers of eggs laid, changes in ovary histology, and altered gene expression [116]. *Caenorhabditis elegans* exposed to E171 showed a concentration-dependent effect on worm reproduction, brood size, and overall display a reduced life span as well as TiO_2_ accumulation in their intestine [117]. E171 exposure to *D. melanogaster* showed an increase in pupation time, changes in the development of larvae, and altered overall reproductive activity, which was accompanied by gene expression changes of CAT and SOD [118,119].

While some meta-analyses report on a publication bias of TiO_2_ toxicity, it is noteworthy that the majority of the conducted studies are performed with nanomaterial models and are not executed according to OECD guidelines, including an insufficient number of test animals, as well as missing particle characterization, or relevant route of exposure.

Some of the publications summarized above show organ-specific toxic effects on the liver, ovaries, and brain, especially in studies conducted with TiO_2_ nanoparticles. These studies reported the onset of inflammation and changes of gene expression related to the immune system, oxidative stress as well as alterations in the oxidant-antioxidant balance system. In vivo studies also show genotoxic effects, including single- and double-strand DNA-damage, micronuclei, and telomere shortening. Other studies, with similar experimental set-up, do not confirm these results, questioning whether the observed effects in vivo do subsequently result in irreversible adverse health effects.

### 5.2. In Vitro and Ex Vivo Toxicity of E171

Table 3 shows an overview of recent publications summarizing the effects of TiO_2_ nanoparticles and E171 on various cell models along the oro-gastrointestinal route, as well as cell types found in organs, following the systemic distribution of these particles. Some studies on TiO_2_ showed the ability to decrease cell viability and induce the formation of ROS, while others do not detect such effects [16,57,58,120,121,122,123,124,125,126,127,128,129,130,131]. In some cases, the increase in ROS was accompanied by elevated oxidative stress levels and lipid peroxidation, which may lead to the induction of DNA damage and micronuclei [16,123,127,129,130]. These events were accompanied by alterations of antioxidant enzymes, such as SOD, GSH, CAT, and glutathione reductase (GR) [121,122,124]. It has been shown that exposure to TiO_2_ can impair cell membrane integrity, decrease mitochondrial membrane potential, and affect tight junctions [57,58,125]. Other publications reported membrane permeabilization, lysosomal dysfunction, and the initiation of autophagic processes, including a decrease in phagocytic rate and index and changes in the gene expression for autophagy proteins 1A/1B-light-chain-3 (LC-3) and Beclin-I [128,132]. Additional alterations on gene expression related to inflammatory pathways including extracellular signaling regulated kinase (ERK 1/2), Akt, as well as tumor and inflammation-related proteins e.g., p53, BAX, Cytochrome-c, Apaf-1, COX-2, transcription factors such as NF-kB, Nuclear factor erythroid 2-related factor 2 (Nrf2) and caspase-3, 9 have been published and suggest the onset of a tumor-like phenotype [120,128,129,133]. The stimulation of inflammatory processes is indicated by increased production and release of pro-inflammatory cytokines such as TNF-⍺ and IL-8 [58].

## 6. Mode of Action

Many in vivo and in vitro studies showed that exposure to TiO_2_ can result in the formation of ROS and the induction of genetic damage and that E171 has the potential to initiate and stimulate inflammation, promote tumors and impair the overall intestinal health through a similar mode of action. The molecular mechanisms potentially leading to the initiation of E171-induced adverse health effects are still under investigation but are presumed to be closely related to the potential of TiO_2_ to induce ROS formation and to promote inflammation. Mechanistic studies in vitro and in vivo, mostly performed with TiO_2_ nanoparticles suggest that the absorption of TiO_2_/E171 can induce the formation of ROS and could lead to a misbalance of the oxidative-antioxidative system. Two postulated initiating events (IE) and several key events (KE) related to adverse effects after TiO_2_ ingestion have been identified. KEs include ROS generation, oxidative stress, persistent inflammation, persistent epithelial injury, increased cell proliferation, and DNA damage, resulting in preneoplastic lesions and ultimately intestinal adenomas/carcinomas as seen in Figure 2 [3,85].

E171 ingestion and its absorption into intestinal enterocytes (IE1) could result in the formation of ROS, due to the semiconducting properties of TiO_2_ that can lead to the formation of radicals and a misbalance of the oxidative-antioxidative system [33,134,135,136,137,138]. The absorption of E171 into the intestinal enterocytes could furthermore lead to the direct induction of inflammation via lysosomal membrane permeabilization and eventually induce direct DNA damage to intestinal epithelial cells [132,139].

The presence of E171 in the intestine possibly alters the composition of the commensal gut microbiota (IE2), which may result in changes in the metabolic function and result in an increase in pathogen-associated-molecular-patterns (PAMPs), such as LPS [85]. The presence of LPS, the internalization of TiO_2_ particles itself, and TiO_2_ related alterations of the adenosine triphosphate (ATP) flux are potential triggers for the activation of the NLPR-3 inflammasome, leading to the subsequent release of IL-18 and IL-1β [85,104,140,141,142,143]. Inflammatory conditions have also been shown in M-cells of the Peyer’s patch. E171 ingestion and absorption led to a decreased T-cell population in Peyer’s patches, which may result in local immunosuppression and pro-inflammatory conditions [14,103]. T-cell receptor suppression associated increase in COX-2 gene expression elevates the production and release of prostaglandin-2 and acts as another mediator for inflammation and GPCR stimulation [120]. Gene expression analysis in vitro and in vivo showed significant alterations in GPCR-family, olfactory receptors, and DNA repair. Molecular pathways relevant for tumor development, including oxidative stress, immune response, inflammation, and cancer signaling showed changes after E171 exposure. Even though the genes affected in these different models were not identical, they showed similarities in their processes. This included increased expression of NF-kB and IRF-3, as well as phosphorylation of tumor suppressor gene p38, along with an increase in MAPK, STAT-1 pathways which are associated with inflammation and cellular stress [44,92]. Additional clustering analysis of RNA-sequences showed the most significantly enriched gene ontology terms and Kyoto Encyclopedia of Genes and Genome (KEGG) pathways relate to Endoplasmic reticulum (ER) stress and initiate an increased cytochrome P450 (CYP450) expression. Such events eventually lead to the activation of MAPK and NF-kB pathways that may further contribute to the induction of inflammation in the intestine after E171 ingestion [88]. Persistent inflammatory processes may weaken the cell membranes and decrease barrier integrity, resulting in impaired barrier function. It is furthermore speculated that general alteration of the gut microbiota (e.g., intestinal dysbiosis) may impair overall intestinal health and increases intestinal permeability [144]. Exposure to E171 can result in a decrease of *TJP-1* gene expression, further weakening the intestinal barrier. These processes potentially increase E171 absorption over time and worsen the conditions or accelerate the adverse health effects [94]. A decrease in intestinal barrier integrity and the consequent increase in E171 absorption elevates the systemic TiO_2_ levels and eventually increases the risk for organ-specific toxicity e.g., neurotoxicity, cardiac toxicity, hepatotoxicity, and reprotoxicity [13,94,108,109,113,145,146,147,148].

Interactions of ROS with DNA and persistent epithelial injury can result in DNA damage and oxidative DNA-base modifications such as 8-Oxoguanine [90,93]. The formation of pre-neoplastic epithelial lesions, intestinal adenomas/carcinomas, or alterations in the size of tumors has been shown to be accompanied by downregulation of epidermal growth factor receptor (EGFR) and AKT, as well as downregulation of tumor suppressor genes e.g., *p53* and *p21* [129,149,150]. An up-and downregulation of *p53* has been observed, which probably displayed different stages within the onset of adverse effects, triggered by E171 exposure. In an earlier stage of adverse effect onset, the upregulation of *p53* tumor suppressor gene initiates an increased binding of p53 to DNA to stimulate p21/cdk2 complex building in case of DNA damage, to stop cell division. E171 also appears to increase *Apaf-1* and *BAX* gene expression [129]. *Apaf-1* is a cytoplasmic protein initiating apoptosis. BAX is an apoptotic regulator, which binds with the *BCL-2* gene to initiate apoptosis and is closely related to the p53 pathway and the formation of colon cancer [129,150]. Upregulation of the master regulator for lysosomal biogenesis and autophagy transcriptional factor EB (TfEB) as well as the increased expression of LC-3 and Beclin-1 (predictive markers for colorectal cancer) contribute to the assumption that the exposure to E171 and the resulting induction of persistent inflammation could be associated with an increased risk for colorectal cancer [128].

As E171 is suspected to impair intestinal barrier functions through dysregulation of inflammatory cytokines, immunosuppression, and alteration of the commensal gut microbiota. It may be expected that patients already suffering from IBD could be a group of people more susceptible to E171 related adverse health outcomes. The examination of blood titanium levels in a swiss IBD cohort support this hypothesis and showed significantly increased blood titanium levels in patients suffering from an active phase of ulcerative colitis and potentially contribute to a worsening of the existing intestinal inflammation [89].

It is currently unclear if E171 is chemically active and a mutagen, or if its mode of action is limited to the formation of free radicals, that promote the growth of tumors. Very limited information regarding the direct interaction of TiO_2_ with DNA is reported, but it is under discussion if TiO_2_ can enter the nucleus. A recent study from Du et al. showed no positive mutagenic response of TiO_2_ nanoparticles in the mouse lymphoma assay nor the Ames test [151]. Though the Ames test is not considered to be suitable for an assessment of the mutagenic potential of nanomaterials, due to the presumed inability of bacterial cells to take up particles [2]. Investigation of human responses to E171 exposure and the comparison of molecular biological processes observed in vivo and in vitro may help to better understand potentially harmful effects following the ingestion of TiO_2_ as a food additive [23].

## 7. Recommendation and Outlook

The experts at the workshop on potential health effects of the food additive titanium dioxide (E171) formulated recommendations for future studies, that aim to reduce uncertainties and fill in existing knowledge gaps [23]. These recommendations have been updated and extended to include recent literature. Good Laboratory Practice (GLP) and following OECD guidelines for in vitro and in vivo studies, including dose-response relationships, adequate controls are a given and will aid in the comparability of studies and the compliance with international scientific standards.

### 7.1. Particle Characterization

The identification of the crystal structure and a precise particle characterization to acquire information about particle size and potential matrix effects are crucial. The characterization of morphological or physiological changes of E171 due to their environment is especially important in both the in vitro and in vivo assessment. Complex biological and physiological cell culture media can contain electrolytes, lipids, or proteins, and vary in their pH values, which potentially alters initial particle characteristics. Examination of the state of aggregation and colloidal stability in a time-based manner, could therefore provide valuable insights into particle behavior, help to better understand particle uptake, and eventually lead to adjustments regarding dosimetry [37,152]. As E171 is brought into commerce in different crystal forms, additional information is needed in which crystal form it is manufactured and how this may affect toxicity. As previously described, the particle size distribution is thought to be a major factor in the toxicity of E171 and can have significant effects on particle properties and reactivity. The TiO_2_ nanoparticles and microparticles can be characterized by Transmission/Scanning Electron Microscopy (T/SEM), Dynamic Light Scattering (DLS), Nanoparticle Tracking Analysis (NTA), Fluorescence Correlation Spectroscopy (FCS), as well as single-particle ICP-MS or high-resolution MS. As the characterization of any nanomaterial, and also TiO_2_ and thus E171, depends on the manufacturing conditions as well as on its immediate environment, E171 should be characterized in its pristine form and other matrices, such as in a vehicle/dispersion before dosing in toxicity studies, in the food matrix, or inside the gastrointestinal environment where protein coronas may be formed. In the characterization process, the hydrodynamic diameter including the state of agglomeration and aggregation should be identified, preferable following standardized protocols to enable that results of various studies are comparable and to reduce confounding effects due to variations in the preparation of the samples. A potential approach for such a standardized protocol was made by Kobayashi et al. (2019) and the Joint Research Center. These protocols include recommendations for sonication time, volume, and appropriate vehicles to achieve long-term stability of the particle liquid-phase dispersion [153,154,155,156]. An ever-growing number of studies, assessing the potentially toxic effects of E171, has been published, that associate exposure to E171 with a variety of adverse health outcomes. The majority of these studies evaluated the hazard caused by nanosized TiO_2_ models (<100 nm). However, to adequately assess the food additive E171 the micro-sized fraction, is also of importance. To assess the risk of E171 via ingestion, it is therefore required to conduct an in-depth characterization of the particle and to compare these test materials with commercially available and used E171 [43,157]. To determine the exact composition of TiO_2_ used as food additive E171 is not an easy task, due to its various interaction and alterations within food matrices [28]. But it is necessary to further investigate E171 and to evaluate products circulating on the market, that are known to contain E171, to give an approximation of the E171 composition that is found in those products.

### 7.2. In Vitro Models

In vitro experiments with E171 should consider potential effects on the state of agglomeration or aggregation of E171 following digestive processes in the mouth cavity, stomach and intestine before exposing cellular systems to E171. Protocols for a successful in vitro digestion have been published and are already in use in the assessment of E171 [158,159]. E171 is presumed to be very persistent, it shows a very low dissolution rate in simulated gastrointestinal fluids and tends to agglomerate over time in the digestive cascade [160,161]. Recent studies nevertheless demonstrated that the digestion of TiO_2_ nanoparticles can have an impact on in vitro testing. The application of the INFOGEST 2.0 in vitro digestion method in Caco-2 and HT29-MTX-E12 cells showed a pronounced increase in cytotoxicity after digestion of certain types of tested TiO_2_ nanoparticles, probably due to slight changes in the nanomaterial characteristics [162]. Therefore, it is advisable to include digestive protocols in the in vitro toxicity assessment of E171, if possible [158,159]. Another improvement in the testing of E171 would be the use of advanced in vitro cell models, such as 3D colon organoids. Traditional 2D cell culture models such as differentiated Caco-2 monolayers are not sufficient to adequately assess transport or adverse effects of E171, due to their limited presence of intestinal cells types like mucus-secreting HT29/MTX cells, which are responsible for the transport of TiO_2_ to a significant degree [163]. Differentiated co-cultures of Caco-2/HT29-MTX mucus-producing cells or Caco-2/Raji B cells are an improvement but still do not mimic the complex microarchitecture of the intestine accordingly [164,165]. The intestine is an important and complex organ, which is not properly represented in common cell culture models, due to their lack of organ-specific microarchitecture and a physiological extracellular matrix microenvironment. The proximity of 3D colon organoid cell models to the human colon, with critical self-renewal and maintenance function, reassembles the intestinal tissue to a higher degree and can provide a more realistic model for in vitro experiments. Although immune intraepithelial cells are lacking in human gut organoids, the origin of 3D colon organoids from induced pluripotent stem cells results in the presence of a large variety of physiological cells, such as intestinal stem cells, enterocytes, enteroendocrine and Paneth cells, goblets (mucus-secreting) cells and mesenchymal cells [166,167,168,169].

### 7.3. Rodent Studies

Future research should include a carcinogenicity study (OECD 451), with well-characterized particles and TiO_2_ reassembling commercially used E171. Whereas 2-year carcinogenicity studies are considered the gold standard in identifying carcinogens, they require many animals to obtain sufficient sensitivity for the detection of adverse health effects with tumor formation as the final endpoint. The development of certain adverse health outcomes, such as colon cancer, in untreated rats, is highly unusual [170]. Such studies may have limitations, as healthy animals for instance may not represent the best model for more susceptible populations with a compromised barrier function and a higher predisposition for developing colon cancer [23]. As reported by Ruiz et al. (2017), IBD patients with an active UC have significantly elevated levels of blood titanium concentration, making this ever-growing group of patients particularly vulnerable to the potentially harmful effects of TiO_2_ [89]. These findings may require having a closer look at potential pre-existing conditions, with additional testing in special animal models, e.g., colitis-induced mouse models. Another issue arising regarding long-term carcinogenicity studies is the route of administration. A physiological administration via the diet closely resembles real-life scenarios but may result in a potential underestimation of the risk, due to insufficient systemic exposure, while administration via oral gavage may lead to a potential overestimation of the risk due to exposure to highly dispersed particles. Additional in vivo studies in rodents should be promoted to confirm effects as they have been identified so far, e.g., oriented towards tumorigenesis and the possible underlying mechanism of immune suppression. Gut-associated lymphoid tissues as well as the intestinal microbiota should be considered as possible factors for (pre)tumor formation. Given the possible adverse health effects of ingesting E171, the trade associations intended to conduct an additional laboratory animal study on demand of EFSA, e.g., an Extended One-Generation Reproductive Toxicity Study (EORGTS), in which cohorts are included for reprotoxicological and immunotoxicological tests. BuRO recommended that this research also includes analyses of parameters that are important for the development of colon cancer, for which indications were found in the previously mentioned studies [23]. Future studies should also provide information on internal dose-response relationships, starting from relevant levels for the general population. Studies to be carried out should include parameters that have been shown to be affected by TiO_2_ or that are important early markers of tumor formation, including putative immune and bacterial factors.

### 7.4. Non-Rodent Studies and Human Intervention Study

So far, in vivo studies have been performed in rodents. To substantiate the relevance of these effects, testing in other species, better-mimicking humans would be advisable. A candidate could be the mini pig, due to its closer resembling of the human cardiovascular system and immunological processes [171,172,173]. Even though the options to test in humans are more limited and will require ethical examination, testing in humans would be highly preferred. It is recommended to evaluate if a human intervention study can be undertaken, that will assess key parameters as identified in the in vitro and animal studies, and in which subjects are exposed to E171 in concentrations that are sufficient to allow effects on the outcome parameters to occur. In analogy with animal studies evaluating gene expression changes, conducting gene expression analysis in humans may be valuable, as these parameters could easily be assessed and may demonstrate the potential activation of crucial processes related to oxidative stress, inflammation, and tumorigenesis under relevant human exposure conditions [23]. It is especially important to also focus on differences in intestinal permeability between humans. Inclusion of IBD patients that might be more prone to TiO_2_ related adverse health effects, could potentially contribute to the identification of adverse health effects or populations most at risk. It is recommended that if such studies be carried out, they should be performed in close collaboration with risk assessors, risk managers, as well as clinicians.

## 8. Summarizing Conclusions of the Workshop

Below, the summarized questions initially raised by the NVWA are answered, based on the workshop discussion and publications released after the workshop. Additional recommendations for future research are outlined to guide the further assessment of potential adverse health effects of ingested E171.

Does oral exposure to E171 reveal a relevant toxicological hazard in in-vitro and in-vivo studies and how reliable are these studies?

Adverse effects were identified including the generation of ROS, alterations of the gut microbiota, persistent inflammation, and other effects on the immune system. These conditions can result in persistent epithelial cell injury and potentially lead to DNA damage and exert a tumor-promoting effect of E171. The findings are inconsistent throughout different species and independent research groups. Some studies are revealing a promotion of colonic precancerous lesions or promotion of tumor formation in mice, using models that involve the chemical induction of colon cancer, while others do not. These models are mainly used as research models and a proper investigation of a dose-response relationship was not performed. With the existing findings, it is not possible to carry out a risk assessment. Currently, in vitro studies are insufficient to form the sole base for risk assessment too. Nevertheless, the results of independent researchers point in a similar direction and should be further investigated.

2.Are the animal models, the exposure conditions, and the effects observed in these studies relevant to humans?

It is not clear to what extent the effects observed in animal models are relevant to humans since it is unknown if the mechanisms that lead to these effects occur in humans. Although laboratory animal models have their limitations, they are still toxicologically relevant. For this reason, the extrapolation of dose-related effects on the colon of rats to humans is considered to be valid. Doses used in laboratory animal studies of 5 and 10 mg/kg bw/day led to pre-neoplastic lesions and 5 mg/kg bw/day already led to the promotion of colon cancer. These doses are in the same order of magnitude as the 95th percentile of 14.8 mg/kg bw/day for children in Europe. Because of the variety of E171 forms and their different physiochemical properties e.g., particle size distribution, the studies involving mice and rats may not represent all forms of E171 on the market. Furthermore, it is not clear whether the absorption of TiO_2_ increases with higher intake, due to its effects on microbiotic health and intestinal barrier integrity. It is therefore important to determine tissue titanium concentrations in relevant organs in both rodents and humans.

3.Can the data from in vitro and in vivo studies with TiO_2_ be extrapolated to humans?

The extrapolation of animal testing data to humans is commonly used in risk assessment. Differences between laboratory animals and humans (interspecies) and between human individuals (intraspecies) are considered by using uncertainty factors. Therefore, it is possible to extrapolate the data from laboratory animals exposed to E171 to humans. However, there is no reliable dose-response data in this case.

4.Are there epidemiological studies on the effects of E171 in humans after oral exposure?

There are very few epidemiological data on the adverse effects of E171 in humans available. Except for a limited number of oral absorption experiments and the observation of increased blood TiO_2_ levels in UC patients from a Swiss IBD cohort. Yet no associations between E171 exposure and the induction inflammatory responses, alteration of gut microbiota, and colon cancer have been reported in humans [68,69,70,89].

## 9. Conclusions

After reviewing the literature on the potential risks of oral exposure to TiO_2_, we concluded that the existing body of evidence raises concern for human health regarding the long-term ingestion of E171. The wide-spread human exposure in combination with the reported tumor-promoting and pro-inflammatory responses in animal experiments indicates the necessity to fill in the identified knowledge gaps that are crucial in the hazard identification and risk assessment process. A particular concern was identified for children due to their proportionally higher TiO_2_ intake, and patients with IBD, given their potential risk of increased absorption, as a consequence of impaired intestinal health.

Animal experiments have shown that chronic exposure to E171 can lead to translocation and bioaccumulation of TiO_2_ via the bloodstream, in various organs, including the liver, kidney, placenta, and brain. Across different types of models, gene expression patterns have been reported that are associated with inflammation and tumor development. In-vivo, ex-vivo, and in-vitro experiments, mainly conducted with TiO_2_ nanoparticles, show that TiO_2_ can result in the formation of ROS, which is associated with the induction of genetic damage, the initiation, and stimulation of inflammation, and the promotion of tumor formation. The endocrine and reprotoxic effects found in rodent studies indicate the need for additional research to reduce uncertainties. These complex interplays of molecular mechanisms involving local persistent inflammation, disturbance of the oxidative–antioxidative balance, immune suppression, apoptosis, changes in microbiotic health, and colon cancer-related pathways need to be further investigated to better understand the molecular biological process, their interaction, and involvement following the chronic exposure to E171.

At the workshop, it was noted that chronic carcinogenicity studies in laboratory animals might have limitations in identifying influences on the incidence of rare tumors, such as colon cancer in rats. For this purpose, specific disease models may provide complementary information. Furthermore, it was concluded that proper characterization of the TiO_2_ particles is crucial for future studies and that the type of crystal form and particle size used, both in the commercially available E171 and in experimental toxicity studies, should be well described. While oral exposure of TiO_2_ via drinking water (oral gavage) and via the diet is both relevant, the effect of the food matrix on bioavailability and adverse health effects is still poorly understood and potentially has an influence on particle properties and toxicokinetics, hence should be considered in the hazard identification of E171. Finally, human dietary intervention studies are needed to demonstrate or discard adverse responses to E171, under relevant exposure conditions, and to better understand the potential cellular and molecular mechanism of action in humans.

## Figures and Tables

**Figure 1 ijms-22-00207-f001:**
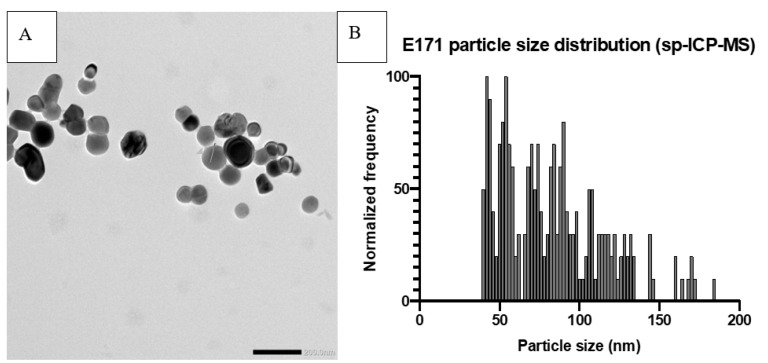
Example of E171 particle characterization. Prior analysis the samples were dispersed according to the NanoGenotox dispersion protocol at a final concentration of 2.56 mg/mL in 0.05% BSA solution and probe sonicated on ice for 16 min (4 W). (**A**) Transmission Electron Microscope picture of E171. (**B**) Size distribution of E171 particles, measured by single-particle ICP-MS, with a median particle size of 79 nm and 72% of particles < 100 nm.

**Figure 2 ijms-22-00207-f002:**
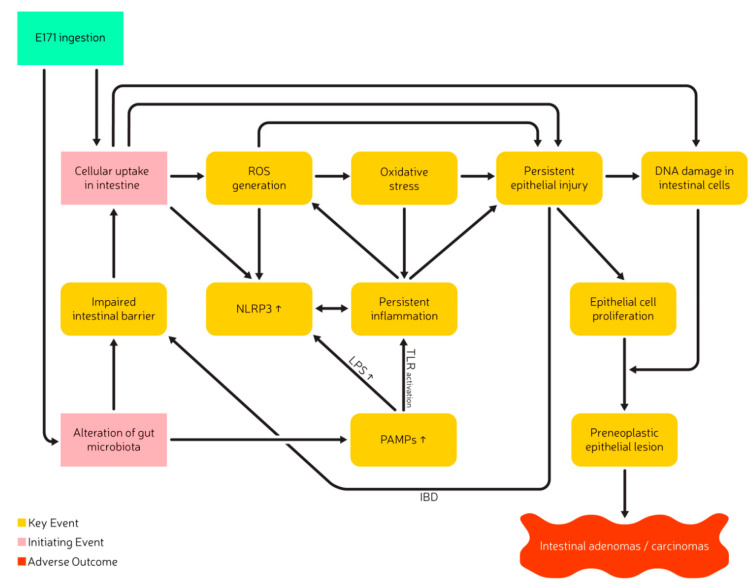
Postulated Adverse Outcome Pathway of TiO_2_ after ingestion and related to inflammation and carcinogenicity, adapted and extended from Braakhuis et al. [3]. Abbreviation: ROS, reactive oxygen species; PAMPs, Pathogen-Associated-Molecular-Patterns; LPS, Lipopolysaccharides; IBD, Inflammatory Bowel Disease; NLRP3, NOD-, LRR- and pyrin domain-containing protein 3, TLR, Toll-like receptor.

**Table 1 ijms-22-00207-t001:** Mean and 95th percentile estimation of daily oral intake of TiO_2_ from food products (E171), food supplements and toothpaste in different age groups and countries in mg/kg bw/day (n/a = data not available, * mg/person/day).

Author	Year	Country	Mean (mg/kg bw/Day)	95th Percentile (mg/kg bw/Day)
Wu [50]	2020	USA	0.15–3.9 * (PCP survey and usage patterns, no food included)	n/a
EFSA [21]	2016	Europe	<11 months: 0.2–0.8 1–3 years: 0.6–4.6 3–9 years: 0.9–5.5 10–17 years: 0.4–4.1 18–64 years: 0.3–4.0 >65 years: 0.2–2.8	<11 months: 0.7–3.9 1–3 years: 2.0–6.8 3–9 years: 2.4–14.8 10–17 years: 1.3–10.8 18–64 years: 1.1–9.7 >65 years: 0.5–7.0
Rompelberg [49]	2016	NL	2–6 years: 0.66–0.70 7–69 years: 0.16–0.18 >69 years: 0.05–0.07	2–6 years: 1.19–1.40 7–69 years: 0.47–0.54 >69 years: 0.20–0.28
Bachler [51]	2015	DE	“Other Children”: ~2 Toddlers, adolescents, adults, elderly: 0.5–1	“Other Children”: ~0.7–7.2 Toddlers, adolescents, adults, elderly: ~0.1–4.2
Sprong [52]	2015	NL	2–6 years: 1.3–1.5 7–69 years: 0.6–0.7 >70 years: 0.5–0.6	2–6 years: 4.5–5.6 7–69 years: 2.6–3.0 >70 years: 1.7–2.2
Christensen [53]	2015	DK	Children: 2 Adults: 1	n/a
Weir [27]	2012	UK	<10 years 2–3 >10 years: 1	n/a
Weir [27]	2012	US	<10 years: 1–2 >10 years: 0.2–0.7	n/a
Powell [54]	2010	UK	5 *	n/a

**Table 2 ijms-22-00207-t002:** Overview of in vivo studies assessing adverse health effects following TiO_2_ ingestion related to acute, sub-chronic and chronic toxicity, genotoxicity, inflammation, histopathological changes and other adverse health outcomes in rats and mice. Abbreviations: BW = bodyweight, ACF = aberrant crypt foci, IS = immune system, OS = oxidative stress, ER = endoplasmic reticulum, A = adult, Y = young, WT = wild-type, * mg/kg/week, ** µg/mL, *** administration from Monday to Friday (5 days a week), **** see original document, due to variety of particles).

Reference	Testing	Material	Ø Primary Size (nm)	Hydrodynamic Diameter (nm)	Zeta Potential (mV)	Species/Sex ♀♂	Duration (days)	Dose (mg/kg bw/day)	Administration	Observation
Chen 2019 [99]	90-days repeated dose	TiO_2_ NP Spherical anatase (purity 99.90%)	29 ± 9	In water ~30 In gastric juices ~105 In intestinal juices ~110	In water ~+10 In gastric juices ~+5 In intestinal juices ~−15	Rats *Sprague-Dawley*	90	0, 2, 10, 50	Oral gavage in ultra-pure water	Slight hepatotoxicity at 50 mg/kg/day including mitochondrial swelling Changes in liver metabolic function Changes in metabolic function of gut microbiota, leading to ↑ LPS ↓ GSH/GSSG ↑ activity of GPx, SOD, MDA ↑IL-1⍺, IL-4, TNF in serum
Moradi 2019 [86]	14-days repeated dose	TiO_2_ NP 80% anatase, 20% rutile	20	Not assessed	Not assessed	Rats ♂ *Wistar*	14	300	Oral gavage in bi-distilled water	Hepatic injury, redox perturbation ↑ serum levels ALT, AST, ALP, LDH ↓ activity GPx, SOD, CAT ↑ mRNA expression NF-kB, TNF-⍺ Histopathological changes in liver e.g., hypertrophy of Kupffer cells
Jensen 2019 [94]	70-day repeated exposure	E171 99.8% anatase, 0.2% rutile	135 ± 6 305 ± 61 900 ± 247	In filtered water, 2% FBS 270 ± 25	Not assessed	Rats ♀ *Zucker*	70	50 *, 500 *	Oral gavage in sterile water 2% FBS	↓ gene expression of *TJP-1* Telomer shortening in lung No changes in oxidative DNA damage No changes in DNA repair activity in liver or lung
Blevins 2019 [17]	100-days repeated dose	E171 85% anatase 25% rutile	110, 115	In food, no assessment of E171 characteristics in food	In food, no assessment of E171 characteristics in food	Rats ♂ *Wistar* (immune response and DMH colon carcinogenesis model)	7100	40, 400, 5000	Ingestion via food in Purina 5002R33 diet	No effects on any immune parameters measured No changes in tissue morphology No changes of dendritic cells in Peyers patches, or cytokine production in jejunum or colon No changes in life measures e.g., BW
Bettini 2017 [14]	100-days repeated dose (E171), 7-days repeater dose (E171 and NM-105)	E171 (45% nanosized by particles number), NM-105 85% anatase 25% rutile	E171: 118 ± 53 NM-105: 22 ± 1	In purified water E171: 373 ± 20 NM-105: 192 ± 2	In purified water E171: −23.9 ± 2.4 NM-105: +5.03 ± 0.02	Rats ♂ *Wistar* (immune response)	7100	10	Oral gavage in purified water	Particles translocation into Peyer’s patches (PP) Decreased Th cells and increased dendritic cell population in PP Intestinal immunosuppression after 7 days, colon microinflammation after 100 days ↑ T-helper (Th) Th1/Th17 systemic inflammatory responses (spleen)
Bettini 2017 [14]	100-days repeated dose	E171,	E171: 118 ± 53	In purified water E171: 373 ± 20	In purified water E171: −23.9 ± 2.4	Rats ♂ *Wistar* (normal wild type and DMH colon carcinogenesis model) Wistar	7100	0.2, 10	Oral gavage in purified water	Initiation and promotion of preneoplastic lesions in the colon at 10 mg/kg bw/day only ↑ total number aberrant crypts in colon ↑ number of large ACF per colon ↑ in cytokines TNF-⍺, IL-8, IL-10
Martins 2017 [18]	45-days repeated dose	TiO_2_ NP	41.99 ± 1.63	Sodium citrate buffer 0.1M, pH 4.5 447.67 ± 6.43	Not assessed	Rats ♂ *Wistar*	45	0.5	Oral gavage in sodium citrate buffer	No sign. changes in redox parameters No ↑ genotoxicity in blood or liver No ↑ in OS
Donner 2016 [67]	OECD 474	TiO_2_ ****	****	****	****	Rats ♂♀ *Sprague-Dawley*	2, 3	500, 1000, 2000	Single dose oral gavage in water	All six TiO_2_ forms negative in vivo genotoxicity testing No ↑ of blood or liver TiO_2_ levels Data suggest no absorption of test materials in GI tract
Warheit 2015 [83]	OECD 407	TiO_2_ 2 types of rutile	173	In 0.01% tetrasodium hexametaphosphate + MilliQ 253	****	Rats ♂ *Sprague-Dawley*	28	24000	Oral gavage in sterile water	No treatment changes in related clinical pathology parameters measured No adverse effects on organ weights Microscopic evidence of TiO_2_ in intestinal lymphoid tissue NOAL = 24,000 mg/kg bw/day
Warheit 2015 [6]	OECD 414	TiO_2_ ****	20–206	****	****	Rats ♀ *Sprague-Dawley*, *Wistar*	14	100, 300, 1000	Oral gavage in sterile water	No substance related mortality No substance related clinical observations No maternal or developmental toxicity or adverse effects on either rat strain NOAEL 1000 mg/kg bw/day
Orazizadeh 2014 [87]	14-days repeated dose	TiO_2_ NP	50–100	Not assessed	Not assessed	Rats ♂ *Wistar*	14	300	Oral gavage in milli-Q water	↑ hepatic level of MDA ↑ plasma levels ALT, AST, ALP ↓ hepatic level of GPx, SOD Histopathological changes to rat liver lobular structure, ↑ inflammatory cells ↑ apoptotic index
Hu 2020 [95]	56-, 182-days repeated dose	TiO_2_ NP	25.37 ± 4.17	In PBS 34.34 ± 6.33	In PBS “negative value”	Mice ♂ *ICR* (Y/A)	56, 182	50	Oral gavage in PBS	↑ xenobiotic biodegradation in liver (Y, A) ER stress in liver and OS in liver and serum (Y) Inflammatory response ↑ activity of MAPK → insulin resistance in liver (Y)
Ali 2019 [35]	5-days repeated dose	TiO_2_ NP	21, 80	Not assessed	Not assessed	Mice ♂ *Swiss-Albino*	5	50, 250, 500	Oral gavage in 0.9% physiological saline solution	Positive correlation with dose increase Histopathological change in the liver ↑ CAT, NO, MDA in liver ↑ in serum AST, ALT ↓ GSH in liver ↑ Chromosomal aberration in mouse bone marrow
Chakrabarti2019 [91]	OECD 408	TiO_2_ NP	58.25± 8.11	Not assessed	Not assessed	Mice ♂♀ *Swiss-Albino*	90	200, 500	“orally” in water	Impaired cell cycle Dose dependent ↑ Comet scores (tail length, DNA in tail) ↑ micronuclei, chromosomal breakage in bone marrow ↑ in serum ALT, AST, ALP
Proquin 2018 [92]	21-days repeated dose	E171	535	In water at 1 mg/mL 316.8 ± 282.4	In water at 1 mg/mL −12.78 ± 0.52	Mice ♂♀ *BALB/c*	2, 7, 14, 21	5	Oral gavage in water	Histopathological alteration/disruption of crypt structure inducing hyperplastic epithelium Gene expression changes olfactory/GPCR receptor family, OS, IS, cancer related genes
Hu 2018 [88]	182-days repeated exposure	TiO_2_ NP	26.42± 7.73	In PBS 42.15 ± 6.71	In PBS “negative values”	Mice *ICR*	182	10, 20, 50, 100, 200	Oral gavage in PBS	ER stress due to ↑ CYP450 expression and ↑ OS ↑ inflammatory responses activated MAPK, NF-kB pathways ↑ plasma glucose levels due to insulin resistance ↑ serum levels of MDA ↓ GSH and SOD in serum and liver
Ruiz 2017 [89]	8-days repeated dose	TiO_2_ rutile	30–50	Not assessed	Not assessed	Mice ♀ *C57BL/6* *NLRP3^−^* (DSS colitis model)	8	50, 500	Oral gavage in water	↑ acute colitis, shorter colon in WT mice in presence of colitis ↑ inflammatory cell infiltration, disruption of mucosal epithelium TiO_2_ accumulation in spleen and liver
Urrutia-Ortega 2016 [15]	77-days repeated dose	E171 (purity > 99%)	382 502 626	In water pH7 300	In water pH7 -30	Mice ♂♀ *BALB/c+* (CAC model)	45 ***	5	Oral gavage in water	Dysplastic alterations in distal colon (BALB/c) ↑ COX-2, Ki67, β-catenin ↑ p65-NF-kB ↓ goblet cell number in distal colon Tumor formation only in (CAC model) ↓ IL-2, TNF-⍺, IFN-γ, IL-10 (CAC model)
Sycheva 2011 [93]	7-days repeated dose	TiO_2_ MP TiO_2_ NP	MP: 160 ± 59.4 NP: 33 ± 16.7	Not assessed	Not assessed	Mice ♂♀ *CBAB6F1*	7	40, 200, 1000	Oral gavage in distilled water	MP ↑ micronuclei and DNA damage in bone-marrow NP ↑ DNA damage in bone-marrow, liver ↑ Mitotic index forestomach and colon epithelia ↑ frequency of spermatids with two and more nuclei
Trouiller 2009 [90]	5-days repeated exposure	TiO_2_ NP 75% anatase, 25% rutile (purity 99.5%)	21	In water 160 ± 5	Not assessed	Mice *C57B1/6Jp^un^/p^un^*	5	60, 120, 300, 600 **	Orally in drinking water	↑ 8-OxoG, γH2AX foci, micronuclei and DNA deletion Moderate inflammatory responses ↑ TNF-⍺, IFN-γ, IL-8

**Table 3 ijms-22-00207-t003:** Overview of in vitro and ex vivo studies assessing molecular biological effects, cytotoxicity, genotoxicity and gene expression changes following TiO_2_ exposure to relevant cellular model system Abbreviations: ROS = reactive oxygen species, OS = oxidative stress, ER = endoplasmic reticulum, * various NP mixtures/sizes (see original publication), ** µg/cm^2^, *** ppm).

Reference	Material	Ø Primary Size (nm)	Hydrodynamic Diameter (nm)	Zeta-Potential (mV)	Cell Type	Duration (h)	Conc. (µg/mL)	Observation
Teubl 2015 [57]	NM-103 rutile dimethicone coated NM-104 rutile glycerol coated	NM-103: 20 NM-104: 20	In PBS NM-103: 1819 ± 61.56 NM-104: 1539 ± 489 In artificial saliva NM-103: 3061 ± 134.4 NM-104: 2597 ± 426	In PBS NM-103: −25.2 ± 8.2 NM-104: −21 ± 4.3 Inartificial saliva NM-103: −9.5 ± 3.7 NM-104: −9.1 ± 3.3	Human buccal epithelial TR146	4	1–200	NM-103, NM-104 did not penetrate into mitochondria (LSM) No decrease in cell viability (MTS) ↑ of mitochondrial activity/viability NM-104 decreased mitochondrial membrane potential, resulting in ↑ROS
Teuble 2015 [56]	NM-100 anatase NM-101 anatase NM-105 80% anatase, 20% rutile	NM-100: 148 ± 45/34 ± 15 NM-101: 28 ± 8 NM-105: 36 ± 10	In PBS (d0.5) NM-100: 1346 ± 76 NM-101: 627 ± 5 NM-105: 868 ± 16 In artificial saliva (d0.5) NM-100: 2850 ± 482 NM-101: 1095 ± 28 NM-105: 1183 ± 74	In PBS (d0.5) NM-100: −10.3 ± 8.2 NM-101: −28.7 ± 4.3 NM-105: −24.0 ± 3.4 Inartificial saliva (d0.5) NM-100: −12.0 ± 5.7 NM-101: −8.46 ± 4.8 NM-105: −11.7 ± 5.6	Human buccal epithelial TR146	4, 24	1–200	No effect on cell viability (MTS) No effects membrane integrity (LDH) Small variances in mitochondrial activity ↑ ROS formation for NM-101/-105
Kim 2019 [120]	TiO_2_ NPs anatase 99%anatase	15	n/a	n/a	Periodontal Ligament cells	0.5, 1, 3, 8, 24, 48	2.5–50	Activation of ERK 1/2, Akt, NF-kB signaling ↑ COX-2 gene expression ↑ intracellular ROS formation
Gerloff 2012 [121]	TiO_2_ NPs * (anatase/rutile)	4–215 *	Was assessed *	Was assessed *	Caco-2	4, 24	20, 80 **	Low ↓ cell viability ↓ metabolic activity No effects on ROS formation, DNA damage, GSH levels
Botelho 2014 [133]	E171 80% anatase, 20% rutile	21	In milli-Q water or RPMI supplemented with 10% FBS, or 2% BSA in PBS 420.7	In milli-Q water or RPMI supplemented with 10% FBS, or 2% BSA in PBS −27.8	AGS human gastric epithelial	3, 6, 24	20–150	Tumor like phenotype ↑ cell proliferation ↑ OS, genotoxicity ↓ apoptotic cells
Dorier 2015 [123]	TiO_2_ NPs A12 R20	A12: 12 R20: 20	In water A12: 132 ± 1 R20: >1000 In exposure medium A12: 320 R20: >1000	In water A12: −20,0 ± 0.6 R20: −19.5 ± 0.9 In exposure medium A12: −10.8 ± 0.6 R20: −11.7 ± 0.8	Caco-2	6, 24, 48	50	↑ ROS formation Impairment of redox repair system, without effects on cell viability, DNA damage
Dorier 2017 [122]	A12 E171 * P25 (anatase/rutile)	A12: 12 ± 3 P25: 24 ± 6 E171: 118 ± 53 *	In water A12: 85 ± 2.9 E171 *: 415.6 ± 69.5 P25: 157.6 ± 1.0 In cell culture medium A12: 447.9 ± 0.3 E171 *: 739.3 ± 355.3 P25: 439.9 ± 6.7	In cell culture medium A12: −10.8 ± 0.6 E171 *: −19 ± 0.7 P25: −11.2 ± 0.8	Caco-2/HT29-MTX (co-culture)	6, 48, 3 weeks	0–200	↑ ROS formation ↓ gene expression CAT, SOD, GR ↑ OS after 48h exposure correlating with intracellular accumulation of TiO_2_ No ER stress
Proquin 2017 [16]	E171 TiO_2_ MPs TiO_2_ NPs 99.5% anatase	E171: 50–250 TiO_2_ MPs: 535 TiO_2_ NPs: 10–25	In DMEM 0.05% BSA E171: 669.62 ± 30.13 TiO_2_ MPs: 1385.83 ± 38.85 TiO_2_ NPs: > 1000 HBSS E171: > 1000 TiO_2_ MPs: > 1000 TiO_2_ NPs: > 1000 McCoy + 10% FBS at 1 mg/mL E171: 316 ± 282.4	In DMEM 0.05% BSA E171: −12.97 ± 0.29 TiO_2_ MPs: −14.10 ±0.56 TiO_2_ NPs: −13.12 ± 0.44 HBSS E171: −4.39 ± 0.12 TiO_2_ MPs: −5.62 ± 0.9 TiO_2_ NPs: −6.08 ± 0.09 McCoy +10% FBS at 1 mg/mL E171: −12.56 ± 8.3	Caco-2, HCT116	24	0.001–1000	↑ ROS formation after MP exposure E171, MPs and NPs induced SS-DNA breaks E171 induced chromosome damage (MN) Overall low cytotoxicity
Popp 2018 [132]	TiO_2_ NPs NanoAmor, mkNano MKN-TiO_2_-A100 (purity > 98%)	NanoAmor:15 mkNano: 50 MKN-TiO_2_-A100: 100	In DMEM at 25 µg/mL NanoAmor: ~300 mkNano: ~800 MKN-TiO_2_-A100: ~500	In DMEM at 25 µg/mL NanoAmor: ~−15 mkNano: ~−12 MKN-TiO_2_-A100: ~−12	HeLa	24, 48, 72	0–500	lysosomal dysfunction and membrane permeabilization induction of lysosomal autophagy mediated by TFEB ↑ autophagic efflux
Pedata 2019 [58]	TiO_2_ NPs P25	P25: 21	Not assessed	Not assessed	Caco-2 (differentiated)	4, 24, 48	42, 84	↓ barrier integrity → disruption of TJs ↓ cell viability ↑production of TNF-⍺, IL-8
Dorier 2019 [124]	E171 A12 NM-105	E171: 118 ± 53 A12: 12 ± 3 NM-105: 24 ± 6	In ultrapure sterile water at 10 mg/mL E171:415 ± 69 A12: 85 ± 3 NM-105: 158 ± 1 In DMEM + 10% FBS E171:739 ± 355 A12: 448 ± 1 NM-105: 440 ± 7	In ultrapure sterile water at 10 mg/mL E171: −19 ± 1 A12: −11 ± 1 NM-105: −11 ± 1	Caco-2/HT29-MTX (co-culture)	6, 24, 48	0–200	↑ ROS formation ↓ cell viability No oxidative DNA damages Modulation of GSH level, ↓ SOD
Cao 2019 [125]	E171	E171: 113.4 ± 37.2	Not assessed	Not assessed	Caco-2/HT29-MTX (co-culture), + Raji B	6, 24	10, 150 ***	↑ ROS formation ↓ cell viability ↓ TEER, barrier integrity → ↑ translocation of boscalid, inducing more adverse effects formation
Li 2020 [126]	TiO_2_ NPs	25	Not assessed	Not assessed	HCT116, NCM4660	24	2, 30	Small ↓ cell viability ↑ OS changes in gene expression of microRNA including apoptosis, ↑ cell proliferation, metabolism
Ghosh 2013 [127]	TiO_2_ NPs	100	In PBS 6180.73	Not assessed	Human lymphocyte, human erythrocytes	3	25, 50, 100	↓ cell viability No effect on cell membrane integrity ↑ DNA damage, apoptosis Erythrocytes showed morphological changes
Dai 2019 [128]	TiO_2_ NPs rutile	30–50 50–100	Not assessed	Not assessed	RAW 264.7	24	50, 100, 200, 300, 400	↓ cell viability ↓ phagocytotic rate, phagocytotic index Gene expression changes in mRNA for LC-3, Beclin-I
Brzicova 2019 [131]	TiO_2_ NPs	20–60 *	Was assessed*	Was assessed *	THP-1	24	32, 128, 256	Small ↓ cell viability Ability to reduce cell viability directly relate to NP size
Shukla 2013 [129]	TiO_2_ NPs Anatase (purity 99.7%)	30–70 *	In Milli-Q water 192.5 ± 2.00 In CMEM 124.9 ± 3.20	In Milli-Q water −11.4 ± 0.25 In CMEM −17.6 ± 0.48	HepG2	6, 24, 48	1, 10, 20, 40, 80	↑ oxidative DNA damage, micronuclei ↑ ROS formation, changes in GSH levels, lipid peroxidation ↑ activity of p53, BAX, cytochrome-c, Apaf1, caspase-3, caspase-9
Shi 2015 [130]	TiO_2_ NPs Anatase (purity 99.7%)	10–25	DMEM + 1% FCS 366.5 ± 30	DMEM + 1% FCS −26.93 ± 0.43	HepG2	24	0.1, 1, 10	↑ DNA damage ↑ Nrf2 mRNA and protein expression ↑ ROS, lipid peroxidation

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
