# Peer review of "Possible Adverse Effects of Food Additive E171 (Titanium Dioxide) Related to Particle Specific Human Toxicity, Including the Immune System"

_ijms, 2020, doi:10.3390/ijms22010207_

Round 1
Reviewer 1 Report
Dear authors,
congratulations on your paper, the text is well organized and the tables allow for easy reading of the information. The only entry is in the bibliography which has a double numbering.
Author Response
Reviewer 1
We thank the reviewer for taking the time to asses this manuscript. We have revised the manuscript in accordance to the remarks made and removed the double numbering of the bibliography. Additionally grammar and spelling mistakes have been corrected.
Reviewer 2 Report
This is a very interesting and easy to read review on the potential impact of TiO2-NP on health. The review is well organized and clearly presenting all the concepts including basic ones and I appreciate this.
My two main points are:
- Page 6 line 238-239 : The authors talk about possible bio-accumulation that would be toxic for the accumulation but the percent of entry is very low. Could the authors really make an estimate of the bioaccumulation to get an idea of the possible concentration of TiO2-NP after let say a month of exposure in real life?
- While it is discussed in the section 7, the fate of TiO2-NP in gastro-intestinal tract is not described in details in previous sections. This is really missing in regards of all the other points comprehensively described in the review.
Herein are my minor corrections:
Page 3, line 138 : in ‘rather than it's the nano- and micro-sized fraction’, it’s the should be changed for ‘its’
Page 4, line 181: the word ‘between’ is missing after ‘contain’.
Page 6, line 252: A space is missing after ppb.
Page 29, line 630, ‘reassembling’ should be corrected in ‘ressembling’.
Spaces between number and unit are frequently missing and there is also a number of typos and grammatical mistakes.
Author Response
Reviewer 2
We thank the reviewer for taking time to revise and assess the submitted manuscript. The spelling and grammar mistakes pointed out by the reviewer have been corrected in addition, a general spell and grammar check of the whole manuscript has been done.
1) We agree with the reviewer that an estimation of a potential risk of bioaccumulation would be a valuable addition to this review. However, such an estimation is beyond the scope of our paper and hard to make given the many uncertainties. For this reason we now chose to include data of titanium concentration of post mortem examined organs (kidney, liver, spleen, ileum, jejunum), as published by Heringa (2018) and Brand (2020). This might give a valuable approximation of what concentration can be found in humans. It should be noted that all samples were obtained from elderly people (average age 86 years), which have a low estimated TiO2 intake. Hence, this should be considered a very conservative estimation fo potential TiO2 concentrations in humans organ (line 334-337, page 7).
2) We agree with the reviewers insightful remark and agree that the fate of TiO2 particles in the gastrointestinal tract should be highlighted more. Since this review is focused solely on the effects of ingested TiO2, we now additionally emphasised in the beginning of the chapter of toxicokinetics (line 264, page 6) that this chapter describes the fate of TiO2 along the oro-gastrointestinal route, following its ingestion (line 262-361, page 6-8). This includes potential uptake in enterocytes and accumulation in the Peyer's patch (line 292-295, page 6), as well as an approximation of how much TiO2 passes the gastrointestinal tract without being taken up (approx. 99%) (line 351-355, page 8). Potential alterations of particle characteristics, due to pH or protein corona formation have been emphasised in the chapter on particlecharacteristics (line 153-181, page 3), and recommendations (line 635-679, page 32).
Reviewer 3 Report
In this review, Bischoff et al., have compiled a comprehensive study of the effects of E171 from the analysis of recent works in the scientific literature. It is a really informative work. I would like to only add some general comments:
The authors point out some controversies between the presumed inertness and low toxicity of E171 and recent findings of adverse effects. Under the label of E171, it might be plenty of particles of different sizes, shapes, morphologies, surface conditions, etc. It would informative to have a figure of an "standard" or typical characterization of E171 samples.
As the authors describe, a significant part of E171 consist of submicrometric or even nano-sized particles. It is acknowledged the great importance of morphologies and surface state of nanomaterials in the observed biological effects. Cellular environments and physiological media contain different and higher ionic and molecular compositions than the media in which the material has been prepared. Similarly, there are different redox states and different pHs inside tissues and cellular structures, as well as the presence of nucleophilic species and ionic scavengers. The processes that (nano)(micro)particles undergo in these conditions are diverse and a variety of parameters are involved that alter their bioactivity. In section 2, some of the possible physicochemical modifications in the physiological environment are briefly summarized. I would suggest to stress out the importance of this time-evolution characterization of the materials tested (E171 in this case) in the physiological media to correlate with the observed biological or toxicological effects (see e.g. Chem. Soc. Rev., 2015,44, 6287-6305 or Small 2020, 16, 1907322). In addition, beyond the characterization of the material employed, it will be good to know if any of the works cited perform such characterization of the evolution over time of the E171 inside the cell media or the animal model or at least include a paragraph of the importance of this time-evolution characterization and how this can be one of the factors behind the mentioned controversies with works showing no toxicity of TiO2 based products.
Author Response
Reviewer 3
We thank the reviewer for taking time to revise and assess the submitted manuscript. Additionally to the thoughtful comments of the reviewer we have corrected some grammar and spelling mistakes.
1) We agree with the reviewer that further elaboration on standards of E171 would be helpful. General characteristics of E171 have been described ranging from 60-300 nm with 10-40% of the particles in the nano-sized range (line 124-128, page 3). Additional to the data from Verleysen et al (2020) that analysed pristine E171 samples from commercial suppliers, we now added some more data from E171 extracted from actual food samples, via HNO3 digestion, including their characteristics (line 132-138, page 3).
2) We appreciate the reviewers insightful suggestion to stress the importance of colloidal stability and effects of different cell culture media on the morphological and physiological characteristics of E171. Therefore, we indicated the important of colloidal stability in section two (line 153, page 3) and furthermore added a paragraph regarding this matter in the recommendations (line 634-642, page 28), including the recommended reference of Moore et al (2015).
Reviewer 4 Report
the authors described in detail tre effect of TiO2NPs as food nanoadditives. the manuscript is clear and the tables make the discussion easy to read. Hovewer, in order to improve the effectiveness of the review, I suggest to add some images (or a multiple panel images) regards the toxicology studies in vitro and in vivo. In addition, some spelling and grammar errors should be revised.
Author Response
Reviewer 4
We thank the reviewer for taking time to revise and assess the submitted manuscript. Additionally to the useful remarks of the reviewer we have corrected grammar and spelling mistakes.
1) We agree with the review that some additional images or graphs will make the text more accessible. Hence we added an example of an E171 characterisation conducted by our research collaborators, with the test material will are using (line 195-212, page 4).